# Genetic Influence on Treatment Response in Psoriasis: New Insights into Personalized Medicine

**DOI:** 10.3390/ijms24129850

**Published:** 2023-06-07

**Authors:** Emilio Berna-Rico, Javier Perez-Bootello, Carlota Abbad-Jaime de Aragon, Alvaro Gonzalez-Cantero

**Affiliations:** 1Department of Dermatology, Hospital Universitario Ramón y Cajal, IRYCIS, Colmenar Viejo km 9.100, 28034 Madrid, Spain; jpbootello@gmail.com (J.P.-B.); carlotababbad@gmail.com (C.A.-J.d.A.); 2Faculty of Medicine, Universidad Francisco de Vitoria, 28223 Madrid, Spain

**Keywords:** psoriasis, pharmacogenetics, pharmacogenomics, therapeutics, polymorphisms, toxicity

## Abstract

Psoriasis is a chronic inflammatory disease with an established genetic background. The HLA-Cw*06 allele and different polymorphisms in genes involved in inflammatory responses and keratinocyte proliferation have been associated with the development of the disease. Despite the effectiveness and safety of psoriasis treatment, a significant percentage of patients still do not achieve adequate disease control. Pharmacogenetic and pharmacogenomic studies on how genetic variations affect drug efficacy and toxicity could provide important clues in this respect. This comprehensive review assessed the available evidence for the role that those different genetic variations may play in the response to psoriasis treatment. One hundred fourteen articles were included in this qualitative synthesis. *VDR* gene polymorphisms may influence the response to topical vitamin D analogs and phototherapy. Variations affecting the ABC transporter seem to play a role in methotrexate and cyclosporine outcomes. Several single-nucleotide polymorphisms affecting different genes are involved with anti-TNF-α response modulation (*TNF-α*, *TNFRSF1A*, *TNFRSF1B, TNFAIP3*, *FCGR2A*, *FCGR3A*, *IL-17F*, *IL-17R*, and *IL-23R,* among others) with conflicting results. HLA-Cw*06 has been the most extensively studied allele, although it has only been robustly related to the response to ustekinumab. However, further research is needed to firmly establish the usefulness of these genetic biomarkers in clinical practice.

## 1. Introduction

Psoriasis is an immune-mediated inflammatory disease that is highly prevalent worldwide, affecting approximately 2–3% of the world population [1,2]. According to the latest Global Burden of Disease study, there were 4.622.594 incident cases of psoriasis worldwide in 2019, with higher-income countries and territories having the highest incidence rate per 100,000 people (112.6, 95% uncertainty interval 108.9–116.1) [3]. Different clinical forms have been defined according to the type of lesions observed. The most common form is psoriasis vulgaris, which accounts for approximately 90% of the disease cases. Guttate, inverse, pustular, and erythrodermic psoriasis constitute other phenotypes of psoriasis [1]. Although it has been classically considered a skin disease, it is frequently associated with extracutaneous manifestations, such as psoriatic arthritis, mood disorders, inflammatory bowel disease, asthma, chronic obstructive pulmonary disease, cancer, metabolic syndrome, non-alcoholic fatty liver, cardiovascular disease, among others, reflecting the systemic nature of psoriasis [4,5,6]. Indeed, in addition to severely affecting the quality of life of the patient [7], psoriasis is associated with an increase in all-cause mortality [8,9].

Its etiopathogenesis remains unclear. It is probable that genetic, immunological, and environmental factors may play important roles in its development [10]. In this regard, different genetic variants have been associated with an increased risk of suffering from psoriasis. As with other autoimmune disorders, psoriasis manifests strong associations with human leucocyte antigen (HLA) molecules, which are involved in antigen presentation and help to identify exogenous proteins. Particularly, individuals carrying the HLA-Cw*06 allele (also known as HLA-C*06:02) have a 10–20-fold increased risk of psoriasis [11]. Other genes encoding cytokines such as TNF-α, IL-17, IL-23, or their receptors [12], as well as genes involved in keratinocyte proliferation, extracellular matrix remodeling, and angiogenesis [13,14] have been shown to increase the risk of its development. Disarrangements of both the innate and adaptative cutaneous immune system are the main drivers of the inflammatory cycle found in psoriatic lesions. Although not fully understood, the activation of keratinocytes, macrophages, neutrophils, and especially, plasmocytoid dendritic cells, leads to the secretion of different cytokines, including alpha and beta interferons (IFN-α and IFN-β), interleukin-1 beta (IL-1β), and tumor necrosis factor-α (TNF-α). In this cytokine milieu, myeloid dendritic cells are secondarily activated via Toll-like receptors (TLRs) and produce IL-12 and IL-23, which induce the proliferation of helper T lymphocytes (Th) and their differentiation toward a Th1 and Th17 profile. TNF-α and both Th17 (IL-17 and IL-22) and Th1 (IFN-γ) cytokines activate keratinocytes proliferation and angiogenesis, two key features in psoriasis pathogenesis. IL-17 also mediates the recruitment of neutrophils and its activation, degranulation, and neutrophil extracellular traps (NETs) formation, which contributes to the initial and maintenance phases of psoriasis [15,16].

Determining disease severity in medical practice requires a thorough evaluation of clinical and patient-reported factors. The Psoriasis Area and Severity Index (PASI) and body surface area (BSA) are frequently used metrics that provide objective assessments of disease severity. The absolute PASI and the percentage of improvement from baseline (e.g., PASI90 for a 90% reduction) are also used to evaluate the effectiveness of treatment. In addition to PASI and BSA, clinicians must consider other factors such as the impact on quality of life, clinical presentation, lesion location, and concurrent psoriatic arthritis to determine overall disease severity [17].

Moderate-to-severe psoriasis is defined by a PASI > 10, a BSA > 10, and/or a Dermatology Life Quality Index (DLQI) >10 [17]. Phototherapies or systemic treatments, such as conventional systemic drugs (methotrexate, acitretin, or cyclosporine) or small molecules (apremilast), are usually the first-line treatments for these patients, with biologics drugs used in cases of no response or contraindications. Biologics are the most effective treatments currently available [18], targeting the main disease effectors. They are classified according to their target in anti-TNF (infliximab, etanercept, adalimumab, and certolizumab), anti-IL12/23 (ustekinumab), anti-IL17 (secukinumab, ixekizumab, and bimekizumab), anti-IL17R (brodalumab), and anti-IL23 drugs (guselkumab, tildrakizumab, and risankizumab). Despite its high effectiveness, not all patients achieve an acceptable response or sustain it in the long term [19]. Different genetic backgrounds, among others, have been involved in this heterogeneity of response [20]. 

Pharmacogenetic methods analyze the associations between patient responses to certain drugs and variants located in a selection of candidate genes. Most of these variants are single nucleotide polymorphisms (SNPs), which constitute substitutions of one nucleotide that occur in at least 1% of the population. Other variants far less frequently addressed are copy number variations (CNVs), which represent a decrease (deletion) or increase (insertions or duplications) in the number of copies of a DNA region. Pharmacogenomics is the field of genetics concerned with the identification of all human genes and the RNA and proteins encoded by them. In this regard, genome-wide association studies (GWASs) allow for an unbiased approach to examining the impact of genetic variants on the drug response by simultaneously testing hundreds of thousands of common polymorphic sites across many genomes [21]. 

This review aims to update the state of the art of psoriasis studies focusing on the influence of genetic variants on drug effectiveness. The identification of genetic markers of clinical response may aid patient selection for better cost-effective decisions. 

## 2. Literature Search

PubMed and Embase searches were conducted using the terms: (psoriasis OR psoriatic OR psoriasis arthritis OR psoriasiform) AND (treatment OR acitretin OR cyclosporine OR methotrexate OR phototherapy OR biological therapy OR anti-tumor necrosis factor OR infliximab OR adalimumab OR etanercept OR certolizumab OR ustekinumab OR guselkumab OR risankizumab OR tildrakizumab OR secukinumab OR ixekizumab OR brodalumab OR bimekizumab) AND (polymorphism OR pharmacogenetic OR pharmacoepigenetic OR pharmacogenomic) AND (response OR loss of effect OR toxicity) on 17 February 2022. 

Firstly, the titles and abstracts of the articles obtained in the first search were reviewed to assess relevant studies. The search was limited to (1) studies written in English or Spanish; (2) studies addressing the influence of genetic factors on the response to drugs used in psoriasis, including topical, phototherapy, conventional, and biologic therapies; and (3) any type of epidemiological study (meta-analysis, clinical trials, cohort studies, case-control, and cross-sectional studies). Systematic and narrative reviews, guidelines, protocols, conference abstracts, and case reports were excluded. Secondly, the full text of articles that met the inclusion criteria was reviewed. Previous systematic and narrative reviews were reviewed to ensure the accuracy of our search and to manually check their reference lists [16,20,21,22,23]. 

Figure 1 shows the literature search process.

## 3. Therapeutic Options

### 3.1. Topical Therapies

Topical therapy remains a cornerstone in the management of psoriasis. It is considered the first treatment option for milder forms of psoriasis, as well as serving as adjunctive therapy in patients undergoing systemic or biologic treatment. The effect on psoriasis is based on its anti-inflammatory and anti-proliferative capacity. Corticosteroids, vitamin D analogs (calcipotriol and tacalcitol), and calcineurin inhibitors are the most commonly used treatments. Most studies to date have focused on vitamin D analogs.

The rs1544410 (BsmI) polymorphism in the vitamin D receptor (*VDR*) gene did not influence the response to calcipotriol in 92 English patients with psoriasis [24]. However, the same polymorphism (Bb heterozygotic phenotype) predicted a significantly better response to topical tacalcitol in a later Italian study (*n* = 25) [25]. Saeki et al. found that the frequency of the F allele in the rs2228570-*VDR* gene polymorphism was lower in non-responders to tacalcitol compared to controls (47 vs. 64%, *p* < 0.05, Japan) [26]. Halshall et al. investigated other *VDR* gene SNPs, such as A-1012G, rs2228570 (FokI), and rs731236 (TaqI), and found that the A allele of the A-1012G variant and the rs731236-T allele were associated with a better response to calcipotriol (UK, *n* = 205) [27]. Conversely, the rs731236-T allele was associated with partial resistance to calcipotriol in a later Turkish study [28]. Another study on a Turkish population found that rs1544410 (BsmI) and rs7975232 (ApaI) showed no significant associations with response to calcipotriol, but patients with the rs2228570-Ff genotype had worse outcomes, while the rs2228570-ff and rs731236-TT genotypes were associated with a better response [29].

Nonetheless, there are discrepancies in these findings. Zhao et al., in 2015 (*n* = 324, China), found no effect of the SNPs mentioned above on calcipotriol response. Instead, the study found a significant association between a loss of response to this drug and the rs2228570-FF and rs11568820-AA genotypes. In addition, when analyzing the response to calcipotriol in combination with acitretin using the same SNPs, the rs11568820-AA genotype showed an increased response to the drug combination compared to the other SNPs [30].

To sum up, these studies suggest that *VDR* gene SNPs may be associated with the response to treatment with calcipotriol or other vitamin D analogs in psoriasis patients, although the studies are highly heterogeneous and showed scarce reproducibility. Further research is needed to clarify the role of these SNPs in clinical practice. 

### 3.2. Phototherapy

Phototherapy, including narrowband ultraviolet B (NB-UVB) and psoralen ultraviolet A (PUVA), among others, is a treatment modality that is well-established in psoriasis treatment. Phototherapeutic regimens use repeated controlled ultraviolet exposures to alter cutaneous biology, aiming to induce remission of skin diseases [31].

There have been several studies searching for an association between genetic polymorphisms and the response to phototherapy in many different genes and populations: *PPARγ2* [32], *IL-12B*, *IL-17A*, *IL-23R, IL-23A* [33], *HLA-C* [34], *IL-17F* [35], *IL-6* [36], and *MC1R* [37], but all of them showed no effect on response. 

The only associations found were on the *VDR* gene. Lesiak et al. analyzed 50 Polish patients treated with NB-UVB for 20 days and found that those carrying the TaaI/Cdx-2 (rs11568820) AA variant showed a worse response measured using the PASI. Other polymorphisms were analyzed, but no other associations were found [38]. In the study conducted by Ryan et al. (*n* = 93, Ireland), patients homozygous for the C allele in the TaqI (rs731236) *VDR* gene polymorphism, which is associated with decreased VDR activity, had a shorter remission duration after NB-UVB [39]. 

### 3.3. Conventional Systemic Drugs

#### 3.3.1. Methotrexate

Methotrexate is a competitive inhibitor of dihydrofolate reductase, thus decreasing folate cofactors required for the synthesis of nucleic acids. Low-dose methotrexate (<25 mg per week) decreases the proliferation of lymphoid cells, which is considered to be the mechanism by which methotrexate improves psoriasis and psoriatic arthritis [40]. One of the first pharmacogenetic studies to determine the response to methotrexate in patients with psoriasis was conducted by Campalani et al. (UK, *n* = 203), which showed how at least one triplet (3R) in the 5′UTR of *TYMS* was significantly associated with both a poorer response to methotrexate and increased hepatotoxicity [41]. *TYMS* encodes thymidylate synthase, which is involved in pyrimidine synthesis and DNA synthesis/repair. Another study by the same research group (*n* = 374) showed how three variants of the *ABCC1* transporter predicted a better response (PASI75) to this drug: rs35592, rs2238476, and rs28364006, as well as two *ABCG2* SNPs: rs13120400 and rs17731538 [42]. These genes encode ATP-binding cassette (ABC) transporters, which have been involved in multidrug resistance [43]. A study on a South Indian Tamil population found that the HLA-Cw*06-positive allele and rs3761548 of the *FOXP3* gene were independent genetic predictors for the clinical response to methotrexate [44]. Other studies have validated the association between the HLA-Cw*06-positive allele and a greater response to this drug in English [45] and Chinese populations [46]. In this last study, a synergistic effect between the HLA-Cw*06-positive allele and the *ABCB1* rs1045742 SNP was also demonstrated [46]. Conversely, Chen et al. observed a worse response to methotrexate in psoriasis patients with the rs1045642 variant [47]. 

Carriers of the rs1801133-TT genotype of the *MTHFR* gene showed a better response rate (PASI75 and PASI90) to this drug at week 12 than carriers of the CT/CC genotypes and also a higher risk of ALT elevation. MTHFR encodes the methylenetetrahydrofolate reductase (MTHFR) enzyme, which is indirectly inhibited by methotrexate [48]. However, this association has not been replicated in other studies [41,42,49]. 

B7-H4 is located in genomic regions associated with susceptibility to type 1 diabetes. In a single-center, cross-sectional study including 265 Chinese psoriatic patients, carriers of the rs12025144-GG genotype had a higher prevalence of DM and worse response to methotrexate in the subgroup of diabetic patients [50]. Regarding annexin A6 (*ANxA6*), a susceptibility factor for psoriasis, the rs11960458-TT/CT genotype was significantly more likely to be unresponsive to MTX in both short (12 weeks) and long-term (1 year) treatment, whereas the rs960709 and rs13168551 polymorphisms were only associated with short-term efficacy [51]. Other polymorphisms have been associated with better (*IL-17F* rs2397084-T allele [52]; *TNIP1* rs10036748-TT genotype [53]) or worse responses (*GNMT* rs10948059-TT genotype; DNMT3b rs2424913-CT/TT genotype [49]) to methotrexate, although studies with larger sample sizes and on different populations will be needed to consolidate these data.

Lastly, there are studies that have used a pharmacogenomics approach to address this issue. A study that used a whole exon high-throughput sequencing technology to detect the DNA sequence of 22 Chinese patients with psoriasis identified 3 SNPs associated with methotrexate response: the T rs216195 variant of *SMG6* and C rs2285421 of *UPK1A* were associated with better outcomes, while the *IMMT* rs1050301 variant A was associated with a lower PASI75 achievement at week 12 [54]. A GWAS conducted by Zhang et al. on a Chinese cohort of patients with psoriasis (*n* = 333) revealed that the rs4713429 SNP, which has a significant impact on HLA-C expression, was significantly associated with methotrexate response [55]. 

#### 3.3.2. Cyclosporine

Cyclosporine is a cyclic undecapeptide with a potent inhibitory action on T lymphocytes. It remains one of the most effective and rapidly acting treatments currently available for psoriasis. Virtually all the diverse manifestations of this disease can respond. The main side effects are nephrotoxicity and hypertension, which limit its usefulness as a long-term maintenance therapy [40].

Not many gene polymorphisms have been studied regarding cyclosporine response. The most studied gene is the *ABCB1*. Vasilopoulos et al. identified that the T allele on the C3435T polymorphism (rs1045642) was associated with a worse response to cyclosporine in a cohort of 84 Greek patients with psoriasis [56]. Chernov et al. came to the same conclusion regarding the same polymorphism, but they also found that the *ABCB1* C1236T (rs1128503) and G2677T/A (rs2032582) SNPs were significantly associated with a negative response to the cyclosporine therapy in the codominant, dominant, and recessive models (*n* = 168, Russia) [57].

Additionally, Antonatos et al. genotyped 27 SNPs mapped to 22 key protein nodes of the cyclosporine pathway in 200 Greek patients. Single-SNP analyses showed statistically significant associations between the rs12885713-T allele of *CALM1* (*p* = 0.0108) and the rs2874116-G allele of *MALT1* (*p* = 0.0006) genes with a positive response to cyclosporine after correction for multiple comparisons [58].

#### 3.3.3. Acitretin

Acitretin’s mechanism of action is not fully understood. It modulates keratinocyte proliferation and differentiation and also has anti-inflammatory and immunomodulatory effects. There is extensive experience in its use. Its main concern is teratogenicity, which severely limits its use in women of childbearing age [40].

One of the first genes studied in relation to acitretin response was the *VEGFA* gene. Its product, vascular endothelial growth factor (VEGF), induces angiogenesis in different settings, including psoriasis [59]. Young et al. analyzed the -460 SNP (rs833061), which is the most common SNP in the promoter region of the *VEGFA* gene (*n* = 106, United Kingdom). The -460 TT genotype was associated with non-response to oral acitretin, whereas the -460 TC genotype was associated with clearance/significant response to treatment [60]. However, this same polymorphism was subsequently studied by Chen et al. in 131 Chinese patients with psoriasis, who found no influence on acitretin response; they also found no association between different variants of the *EGF* gene and treatment effectiveness [61]. Finally, Bozduman et al. analyzed different *VEFGA* polymorphisms in 100 Turkish patients on acitretin. They found no significant associations except for the +405 G > C SNP (rs2010963), in which a subgroup of four patients with the GG genotype showed a better response [62]. 

In a study involving 105 Chinese patients treated with a combination of acitretin and topical calcipotriol, patients carrying the rs4149056-T allele in *SLCO1B1* and/or the rs2282143-C allele in *SLC22A1* showed a worse response based on PASI50 attainment at week 8 [63].

The relationship between mutations in certain interleukin genes and the response to acitretin has also been studied. Lin et al. conducted a study that included an acitretin-treated subgroup of 24 Chinese patients with the TG genotype in the rs3212227 SNP of the *IL12B* gene; these patients had a significantly better response (PASI50) to acitretin than patients with the GG genotype. Additionally, 19 secondary non-responders to anti-TNF-alpha treated with acitretin were included. In this setting, the presence of the rs112009032-AA genotype in the *IL23R* gene predicted a higher PASI75 achievement [64]. 

Borghi et al. studied the effect of a 14-base pair (bp) sequence insertion/deletion (INS/DEL) polymorphism in the *HLA-G* gene in psoriasis patients from Italy treated with acitretin (*n* = 21), cyclosporine, and anti-TNF-alfa. The investigators found a significantly higher frequency of the *HLA-G* DEL allele among the responders (PASI75 at week 16) in the acitretin group [65]. Zhou et al. also evaluated the influence of *HLA* gene variants on acitretin response (*n* = 100, China). After 8 weeks of treatment, the HLA-DQA1*0201 and HLA-DQB1*0202 alleles were independently associated with a better response to the drug [66]. The same research group conducted a study in which whole exome sequencing was performed on 116 Chinese patients. The *CRB2* rs1105223 CC (OR = 4.10, *p* = 0.007) and *ANKLE1* rs11086065 AG/GG (OR = 2.76, *p* = 0.003) genotypes were associated with no response to acitretin after 8-week treatment. Conversely, the *ARHGEF3* rs3821414 CT/CC (OR = 0.25, *p* = 0.006) and *SFRP4* rs1802073 GG/GT (OR =2.40, *p* = 0.011) genotypes were associated with a higher response rate [67].

Finally, other studies have evaluated the association between acitretin and SNPs on *ApoE* [68] and *IL36RN* [69] genes, but these did not show any effects on the response.

### 3.4. Small Molecules

#### Apremilast

Apremilast inhibits phosphodiesterase-4, which increases intracellular levels of cyclic adenosine monophosphate and subsequently downregulates inflammatory responses involving the Th1 and Th17 pathways [22,70]. To our knowledge, only the study by Verbenko et al. assessed the influence of a subset of 78 pre-selected SNPs strongly associated with psoriasis or psoriatic arthritis in apremilast effectiveness. Thirty-four Russian patients were included. Patients carrying the minor alleles rs1143633 (IL-1β), rs20541(*IL-4/IL-13*), rs2201841(*IL-23R*) and rs1800629(*TNF-α*) showed a higher PASI75 achievement [71]. 

### 3.5. Biologics

#### 3.5.1. Anti-TNF-α Drugs

TNF-α inhibitors were the first biologic drugs available for psoriasis treatment. Etanercept acts as a soluble form of the TNF-α receptor and binds to both TNF-α and TNF-β, depleting these molecules. Conversely, infliximab and adalimumab are both monoclonal antibodies that directly target TNF-α. These molecules have been proven to be effective and safe in the treatment of moderate-to-severe psoriasis in randomized clinical trials controlled with a placebo and with conventional drugs [72,73,74,75,76]. Given their cost-effectiveness, for several authors, they may constitute the first line of biological treatment for the disease [77,78]. 

The influence of the HLA-Cw*06 allele in anti-TNF-α response has been extensively studied. In a recent study by Coto-Segura et al. (*n* = 169), HLA-Cw*06 positive patients showed a better response to adalimumab compared to those without the abovementioned allele (OR 2.35, *p* = 0.018) [79]. A previous study by the same Spanish research group (*n* = 116) observed that psoriasis patients who were carrying the HLA-Cw*06 allele together with the insertion-genotype of the two late cornified envelope genes (*LCE3B_LCE3C*) showed a higher probability of reaching PASI75 under anti-TNF-alfa at week 24 (OR 3.14, 95% CI 1.07–9.24, *p* = 0.034). However, when HLA-Cw*06 was assessed independently, statistical significance was not reached [80]. On the other hand, Gallo et al. (*n* = 109) observed that HLA-Cw*06 carriers were less likely to respond to adalimumab, infliximab, and etanercept than HLA-Cw*06-negative patients [81], and these results are in line with those found by Dand et al. in a study that included 839 patients treated with adalimumab and 487 treated with ustekinumab. A multivariable regression model was individually performed on each group of treatment: HLA-Cw*06 was associated with a worse response to adalimumab, defined as a failure to reach the PASI90 at 6 months (OR 0.54, *p* = 1.67 × 10^−4^) [82]. Similarly, Van den Reek et al. found a poorer response to adalimumab (PASI decline at 3 months) in the HLA-Cw*06-homozygous group of patients [83]. To add to this controversy, the HLA-Cw*06 status was not predictive of anti-TNF-α response in a study conducted in British and Irish patients (*n* = 138) [84], as well as in many other studies carried out in Spanish, Italian, and Chinese populations [85,86,87,88,89]. 

Apart from HLA-Cw*06, other HLA variants have also been analyzed. In a cohort of Greek patients with moderate-to-severe psoriasis (*n* = 228), the rs10484554 polymorphism in the *HLA-C* gene was associated with a better response to anti-TNF-α, as well as the *HLA-A*-rs610604 polymorphism with a better response to adalimumab [90]. The *HLA-B/MICA* rs13437088 polymorphism, which has been associated with early onset psoriasis [91], predicted a better response to etanercept in a study involving 81 Spanish patients [92]. Guarene et al. studied the effect of the *HLA-B* Bw4-80I and *HLA-A* Bw4-80I alleles on 48 Italian patients under biologic treatment including infliximab, etanercept, adalimumab, and ustekinumab. The abovementioned alleles present a greater binding affinity to killer-cell immunoglobulin-like receptors (KIRs), which modulate natural killer (NK) cell function. A significantly better response to etanercept was observed in the carriers of the *HLA-A* Bw80I allele [93]. The HLA-B*46 haplotype was not associated with biologic response in a study involving 74 Chinese patients (45 on etanercept) [85]. Finally, the *HLA-G* 14-base-pair insertion/deletion polymorphism (rs66554220) did not modify the anti-TNF-α response in a small group (*n* = 11) of Italian patients [65]. 

Given that TNF-α is the target of the drugs reviewed in this section (etanercept, adalimumab, and infliximab), several authors have raised the possibility that certain polymorphisms in the *TNF-α* gene may be biomarkers for the response to these therapies [21]. A meta-analysis by Song et al. explored the association between *TNF-α* gene polymorphisms and anti-TNF-α response in patients with autoimmune diseases including psoriasis. The analysis included 10 articles and 887 Caucasian and Asian patients. The *TNF-α* -238 (rs361525) G allele, the *TNF-α* -308 (rs1800629) G allele, and the *TNF-α* -857 (rs1799724) C allele were associated with a better response to these drugs. When stratifying by disease type, the *TNF-α* -857 C allele predicted a better response in psoriasis patients (OR = 2.238, 95% CI 1.319–3.790) [94]. However, only 2 studies with a total of 177 Caucasian patients were included [95,96]. In a Spanish study (*n* = 109), the *TNF-α* -238 G allele also predicted a better response to these drugs [81]. However, it was the *TNF-α* -857 T allele, not the C allele, that was associated with a better response to these treatments, and no association between the *TNF-α* -308 polymorphism and anti-TNF-α response was found. Patients carrying the TT genotype of the *TNF-α* rs1799964 polymorphism also showed a better response to anti-TNF-α both at 3 and 6 months [81]. However, the studies by Dapra et al. [97] and Ovejero-Benito et al. [98] found no effect of the four abovementioned polymorphisms on the response to etanercept in Italian and Spanish patients, respectively. A study including 49 Japanese patients with moderate-severe psoriasis also found no effect of the *TNF-α* -857 polymorphism on the response to adalimumab or infliximab [99].

Polymorphisms in other genes related to TNF-α signaling have also been studied. The tumor necrosis factor receptor superfamily member 1A (*TNFRSF1A*) and 1B (*TNFRSF1B*) genes encode two receptors (TNFRI/p55 and TNFRII/p75, respectively) that mediate the signaling of TNF-α. TNFRI mainly mediates inflammatory and pro-apoptotic responses, while TNFRII has been more implicated in immune regulation and tissue regeneration [100]. Chen et al. conducted a meta-analysis to investigate whether certain polymorphisms in these genes could predict the response to anti-TNF-α therapies in patients with autoimmune diseases (rheumatoid arthritis, psoriasis, and Crohn’s disease). The analysis included 8 studies involving 929 subjects with the *TNFRSF1B* rs1061622 polymorphism and 564 subjects with the *TNFRSF1A* rs767455 polymorphism. Only 2 of the included studies were conducted on patients with psoriasis (a Greek and a Spanish cohort with 90 and 80 patients, respectively) [95,101]. Carriers of the rs1061622-T allele showed a better response to anti-TNF-α drugs (OR 0.62, 95% CI 0.40–0.97 for non-response T vs. G). When a subgroup analysis was conducted according to disease type, this association was maintained in the models for psoriasis (OR = 0.39, 95% CI 0.23–0.67) [102]. The influence of this polymorphism could be greater for Etanercept, as shown in the study by Vasilopoulos et al. [95]. 

The tumor necrosis factor-alpha-induced protein 3 (*TNFAIP3*) gene encodes a critical protein that functions as a negative regulator of the NF-κB signaling pathway, which is essential for the activation of the immune response. It is also involved in TNF-mediated apoptosis [16]. The influence of the *TNFAIP3* rs610604 and rs2230926 SNPs in the response to anti-TNF-α drugs was studied by Tejasvi et al. A total of 632 patients with psoriasis from the USA and Canada were included. The efficacy was measured with a visual scale. Patients that carried the rs610604-G allele responded better than those carrying the A allele (OR 1.50, *p* = 0.05). Stratifying by treatment, this difference was only significant for Etanercept (OR 1.64, *p* = 0.016). Although no association was found between treatment response and the rs2230926 SNPs, those who carried the rs2230926 T-rs610604 G haplotype showed better outcomes [103]. In a Spanish study that included 20 patients with psoriasis and psoriatic arthritis, both the AA-rs6920220 and the AC/CC-rs610604 genotypes were associated with a greater quality of life improvement after anti-TNF-α [98]. Conversely, Masouri et al. found that the A-allele and not the C-allele in the *TNFAIP3* rs610604 polymorphism was associated with a better response, which was only significant with Etanercept [90]. These findings are in line with those of an Iraqi study involving 100 patients with psoriasis [104]. Other authors have found no effect of *TNFAIP3* SNPs in response to these drugs [83,99]. 

Alongside TNF-α and related molecules genes polymorphisms, other potential biomarkers have been explored to predict anti-TNF-α response. The Fc fragment of IgG receptors IIA(*FCGR2A*) and IIIA(*FCGR3A*) are surface receptors that bind to the constant fraction of immunoglobulin G (IgG) and participate in antibody-dependent cellular toxicity mediated by phagocytic or cytotoxic cells. Genetic alterations in these genes can affect the receptor’s affinity for the immune complex [16]. The presence of histidine (H) instead of asparagine (R) at position 131 (rs1801274) in *FCGR2A* and valine (V) instead of phenylalanine (F) at position 175 (rs396992) in *FCGR3A* results in higher-affinity receptors. In a Spanish study involving 70 patients with moderate-to-severe psoriasis, individuals carrying high-affinity genotypes (HH131 + HR131 and VV158 + VF158) showed a greater reduction in BSA at week 6 of anti-TNF-α treatment (beta = 0.372, *p* = 0.3 and beta = 0.425, *p* = 0.02, respectively). However, no significant differences were observed at week 12 or in terms of the PASI [105]. Another Spanish study involving 133 patients found that those harboring the low-affinity allele *FCGR2A* were 13.32 times more likely to be non-responders (PASI75 at week 6) to anti-TNF-α drugs [106]. Conversely, neither the study by Mendrinou et al. (*n* = 100, Greece) [107] nor the study by Batalla et al. (*n* = 115, Spain) [108] found an association between the *FCGR2A*-H131R polymorphism and anti-TNF-α response. Moreover, their results for the *FCGR3A*-V158F polymorphism were contradictory: while in the study by Mendrinou et al., the carriers of the high-affinity allele had a better response, especially to Etanercept [107], in the study of Batalla et al., those carrying the lower affinity allele had a greater response, which was only significant in the Etanercept subgroup [108]. 

Antonatos et al. recently conducted a meta-analysis addressing the role of these polymorphisms in the response to anti-TNF-α drugs. It included 37 papers with a total of 8398 Caucasian and Asian patients (4723 diagnosed with rheumatoid arthritis and/or spondyloarthritis, 780 with psoriasis, and 2895 with inflammatory bowel disease). No association was found between the *FCGR2A*-R131H SNP and response to anti-TNF-α overall and when stratified by disease (OR 0.959; 95% CI 0.46–2.02 in psoriasis), which are in line with the results for *FCGR3A*-V158F polymorphisms [109]. The authors also explored *TNF-alfa*, *TNFRSF1A*, *TNFRSD1B*, *TLR1*, *TLR5*, *IL12B*, *IL17A*, and *TRAILR1* SNPs. The results regarding *TNF-alpha* polymorphisms were analogous to those previously reported in the meta-analysis by Song et al. [94] and in the study by De Simone et al. [96]. The T-allele of *TNFRSF1B* rs1061622 was associated with the response to anti-TNF-α in the psoriasis subgroup (2 studies, *n* = 162, OR: 2.62, 95% CI 1.52–4.51). Null results were reported for the rest of the polymorphisms [109]. 

Other SNPs in *IL-17F*, *IL-17R*, *IL-23R*, *IL-12B*, *IL-1*, *IL-6*, *NFKBIZ*, and *CARD14* genes, among others, have been shown to modulate anti-TNF-α response in different settings. Table 1 summarizes them. 

To date, most studies addressing this issue have used a candidate-gene approach, analyzing a limited number of genes previously linked to psoriasis or its treatments. A pharmacogenomic approach base on GWAS, on the other hand, has been used far less frequently. To our knowledge, only three studies were conducted following this methodology [110,111,112]. None of these studies were able to identify any SNP that reached genome-wide statistical significance (*p* < 5 × 10^−8^). Lowering the significance threshold to *p* < 5 × 10^−5^, Nishikawa et al. (*n* = 65) [110] found ten SNPs located in *SPEN, JAG2, MACC1, GUCY1B3, PDE6A, CDH23, SHOC2, LOC728724, ADRA2A*, and *KCNIP1* genes related to anti-TNF-α response; Ovejero-Benito et al. (*n* = 243) [111] identified nine polymorphisms that involved *AKAP13, SUPT3H, CDH12* (2), and *HNRNPKP3* (5) genes; and Ren et al. (*n* = 209) [112] found seven loci associated with the treatment response in the following genes: *IQGAP2-F2RL2, SDC3, IRF1-AS1, NPAP1, KRT31, CTSZ*, and *CNOT11*. Additionally, the authors of the latest work checked the associations for the 19 SNPs with *p* < 5 × 10^−5^ found in the 2 previous GWAS on anti-TNF-α. While two of these SNPs reached *p* < 0.05, none reached the significance thresholds required for these studies [112]. The different genetic backgrounds of the populations assessed in these studies (Japanese, Spanish, and Chinese, respectively) could be behind this lack of consistency. Anyway, GWAS requires a sample size of >1000 patients, so further studies with larger cohorts of patients will be needed in the coming years to validate these results.

In patients with sustained positive outcomes, a common clinical approach is to decrease the dosage or extend the time between administrations (off-label optimization) to minimize side effects and treatment costs. However, this de-intensification poses the risk of treatment loss. Ovejero-Benito et al. discovered that the rs1008953 *SDC4* SNP was linked to successful dose reduction without compromising the response, while certain polymorphisms in *IL28RA*, *TLR10*, *TRAF3IP2*, and *MICA-A9* predicted an inability to achieve it [113]. 

Finally, apart from predicting treatment efficacy, different genetic polymorphisms are involved with toxicity development, especially paradoxical psoriasis (PP). Bucalo et al. explored SNPs in the HLA-Cw*06, *IL23R*, *TNF-α*, and *IFIH1* genes in patients with inflammatory bowel disease or psoriasis and PP under anti-TNF-α. Although they found associations between PP and two SNPs in TNF-α and HLA-Cw*06 in patients with inflammatory bowel disease, no associations were found in patients with psoriasis [114]. Conversely, in the study by Cabaleiro et al. (*n* = 161, Spain), the following SNPs were associated with PP development in psoriasis patients under anti-TNF-α: the rs11209026 *IL23R*, rs10782001 *FBXL19*, rs3087243 *CTLA4*, rs651630 *SLC2A8*, and rs1800453 *TAP1* genes [115]. CNVs in regions involving the *ARNT2*, *LOC101929586,* and *MIR5572* genes have also been associated with the development of PP under infliximab or etanercept in a Spanish study (*n* = 70) [116]. 

#### 3.5.2. Ustekinumab

Ustekinumab is a human monoclonal antibody directed at the common p40 subunit of IL-12 and IL-23, thereby blocking the Th1 and Th17 inflammatory pathways. It has been shown to be effective and safe for the short- and long-term treatment of psoriasis in both clinical trials and real-life studies [111,117,118,119]. 

In contrast to anti-TNF drugs, the influence of HLA-Cw*06 status on treatment response seems to be more consistent for ustekinumab. Talamonti et al. conducted a study that involved 51 Italian patients with moderate-to-severe psoriasis. The authors found striking differences in the rate of ustekinumab response between HLA-Cw*06-positive and -negative patients. A higher percentage of patients achieved PASI75 at week 12 in the HLA-Cw*06-positive group (96.4% vs. 65.2%, OR = 13.4). Carriers of the HLA-Cw*06 also showed a faster response at week 4 and longer disease control [120]. These results were replicated by the same research group in two larger cohorts. In the first one (*n* = 134), a significantly higher percentage of HLA-Cw*06-positive patients reached PASI75 at week 12 (82.9% vs. 54.2%), at week 52 (83.9% vs. 58.2%), and at week 104 (83.9% vs. 60.5%) [121]. A higher percentage of HLA-Cw*06-positive patients achieved not only PASI75 but also PASI90 and PASI100 at weeks 12, 28, 40, and 52 in the second study (*n* = 255) [122]. Taking together, HLA-Cw*06 may predispose to a better, faster, and longer-lasting response to ustekinumab in Caucasian patients. 

Results from other investigations in both Asian and Caucasian populations have found results along the same lines. Chiu et al. studied the influence of different HLA-B and -C polymorphisms in a group of Chinese patients with psoriasis on biologic treatment, including ustekinumab (*n* = 29). Neither HLA-C*01, HLA-C*06, nor HLA-B*46 alleles were associated with a better response in the ustekinumab subgroup (PASI50 at week 12) [85]. However, in a later study by the same investigators (*n* = 66), HLA-Cw*06-positive patients showed a significantly higher mean PASI improvement (81.7% vs. 59.7%) and a higher achievement of PASI90 (62% vs. 26%) at week 28 [123], probably reflecting a lack of power in the former work. Additionally, in the previously discussed study by Dand et al. (*n* = 487 on ustekinumab), patients harboring HLA-Cw*06 were more likely to reach PASI90 at 6 months compared to HLA-Cw*06-negative patients (OR = 1.72, *p* = 0.018) [82]. HLA-Cw*06 was again a predictor of the response to ustekinumab at weeks 4, 28, 40, and 52 in a cohort of 64 Italian patients [124]. 

However, in the study conducted by Raposo et al. (*n* = 116, Portugal), the initial better response to ustekinumab in the HLA-Cw*06 patient subgroup (weeks 12 and 24) was lost at 52 weeks, raising doubts about the consistency of this biomarker across all time points [125]. Another US study determined the HLA-Cw*06 status from 601 participants in three phase III randomized clinical trials [117,118,126], 332 of whom received ustekinumab. A significantly higher percentage of HLA-Cw*06-positive patients achieved PASI75 at week 12 compared to HLA-Cw*06-negative patients (80.6% vs. 62.7%), but the association was again no longer significant in the long-term. Furthermore, there was no strong association between HLA-Cw*06 and the ustekinumab optimal response (PASI90 and PASI100), and the differences between the overall population and the HLA-Cw*06 positive subgroup were minimal (10% or less). In view of these findings, the authors suggested that HLA-Cw*06 status determination may have limited clinical utility in daily practice [127]. Van Vugt et al. outlined a similar conclusion in a meta-analysis that combined most of these individual studies. A total of 8 papers that involved 937 Caucasian and Asian patients were included for the primary analysis (PASI75 at 6 months). HLA-Cw*06-positive patients presented a higher response rate (OR 0.24, 95% CI 0.14–0.35). Indeed, the response rate in the HLA-Cw*06-positive group varied from 62% to 98% (median 92%), whereas it varied from 40% to 84% (median 67%) in the HLA-Cw*06-negative group. Nonetheless, in the authors’ opinion, the actual clinical relevance of HLA genotyping might be questionable as the response rates in both groups were high [128]. Summarizing the evidence available, despite some studies reporting no association [83,90,129], most of the currently available evidence supports the positive influence of HLA-Cw*06 on ustekinumab response. However, further research is necessary to clarify the true clinical impact of HLA genotyping and its potential application in personalized medicine. 

Alongside HLA-Cw*06, other gene polymorphisms have also been studied in relation to ustekinumab. A total of 62 SNPs in 44 different genes were evaluated in a Danish cohort of patients with psoriasis under biologic treatment, 230 with ustekinumab. Two variant alleles of the *IL1B* gene (rs1143623 and rs1143627, OR = 0.25 and OR = 0.24, respectively), which convey increased IL1B transcription, were associated with a significantly lower reduction in the PASI at 3 months of treatment. Conversely, *TLR5* rs5744174 and *TIRAP* rs8177374 polymorphisms, which are genetic variants related to an increased level of IFN-gamma, were associated with a better response (OR = 5.26 and OR = 9.42, respectively) [130]. Van den Reek et al. conducted a study on a Dutch population that included 66 episodes of ustekinumab treatment. The *IL12B* rs3213094-T allele was associated with a greater mean PASI reduction at 3 months, whereas the *TNFAIP3* rs610604-G allele predicted a worse outcome, with an even worse response in psoriatic arthritis patients [83]. Nonetheless, other studies have not been able to replicate the influence of *TNFAIP3* [90,120,124] or *IL12B* polymorphisms [125] in drug efficacy. *IL17-F* [131], *ERAP-1* [90], *CHUK, C17orf51, ZNF816A, STAT4, SLC22A4*, *Corf72, AGBL4, HTR2A, NFKB1A, ADAM33*, and *IL13* [132] polymorphisms have also been associated with the modulation of ustekinumab response. Table 2 summarizes the effect of these SNPs. 

**Table 1 ijms-24-09850-t001:** Pharmacogenetic and pharmacogenomic studies on anti-TNF-α.

Author	Year	Country	Drug	Gen	SNP (Allele/Genotype)	Responsive Allele or Genotype	*n*	Follow-Up	Outcome	Response
**Nani et al.** [133]	2023	Greece	IFX, ADA and ETN	MIR155	rs767649	A	100	24 weeks	PASI75	(+)
**Ren et al. ^γ^** [112]	2022	China	ETN ± MTX	IQGAP2-F2RL2	rs2431355	T	209	12 and 24 weeks	PASI75	(+)
				SDC3	rs11801616	G				(−)
				IRF1-AS1	rs13166823	G				(−)
				NPAP1	rs10220768	C				(+)
				KRT31	rs4796752	C				(+)
				CTSZ	rs4796752	T				(−)
				CNOT11	rs3754679	G				(−)
**Sanz-Garcia et al.** [116]	2021	Spain	IFX, ADA, and ETN	CPM	CVN	-	70	24 weeks	PASI90	(+) ^δ^
**Ovejero-Benito et al. ^γ^** [111]	2020	Spain	IFX, ADA, and ETN	AKAP13	rs28461892	A	243	3 months	PASI75	(+)
				SUPT3H	rs9472377	G				(+)
				CDH12	rs1487419	A				(+)
					rs77497886	T				(+)
				HNRNPKP3	rs11037360	A				(+)
					rs7481533	C				(+)
					rs11037342	C				(+)
					rs145304743	T				(+)
					rs1845821	C				(+)
**Hassan Hadi et al.** [104]	2020	Iraq	ETN	TNFAIP3	rs610604	C	100	6 months	Not specified	(−)
**Coto-Segura et al.** [79]	2019	Spain	ADA	NFKBIZ	rs3217713 (indel)	Ins/Del and Del/Del	169	24 weeks	PASI75	(+)
				HLA-C	HLA-Cw*06	Positive				(+)
**Ovejero-Benito et al.** [98]	2019	Spain	anti-TNF	TNFAIP3	rs610604	AC/CC	20	3 months	EQ-VAS	(+)
					rs6920220	AA				(+)
**Dand et al.** [82]	2019	United Kingdom and Ireland	ADA	HLA-C	HLA-Cw*06	Positive	1326 (839 ADA and 487 ustekinumab)	6 months	PASI90	(−)
**Guarene et al.** [93]	2018	Italy	IFX, ADA, ETN, and UTK	HLA-A	HLA-A Bw480I	Positive	48	6 months	PASI75	(−)
**Ovejero-Benito et al.** [134]	2018	Spain	IFX and ADA	IVL	rs6661932	CT-TT	95	3 months	PASI75	(−)
				IL-12B	rs2546890	AG-AA				(+)
				NFKBIA	rs2145623	CG-GG				(−)
				ZNF816A	rs9304742	CT-CC				(+)
				SLC9A8	rs645544	GG				(−)
**Batalla et al.** [135]	2018	Spain	IFX, ADA, and ETN	IL17RA	rs4819554	A	238	24 weeks	PASI75	(+)
**Prieto-Pérez et al.** [106]	2018	Spain	IFX, ADA, and ETN	PGLYRP-4-24	rs2916205	AG/GG	144	3 months	PASI75	(−)
				ZNF816A	rs9304742	CC		3 months		(−)
				CTNNA2	rs11126740	AA		3 months		(−)
				IL-12B	rs2546890	AG/GG		3 months		(−)
								6 months		(−)
				MAP3K1	rs96844	CT/CC		3 months		(+)
								6 months		(+)
				HLA-C	rs12191877	CT/TT		3 months		(+)
				FCGR2A	rs1801274	CT/CC		6 months		(−)
				HTR2A	rs6311	CT/TT		6 months		(−)
				CDKAL1	rs6908425	CT/TT		6 months		(+)
**Loft et al.** [130]	2018	Denmark	IFX, ADA, and ETN	IL-1B	rs1143623	G/C	376	3 months	PASI75	(−)
					rs1143627	T/C				(−)
				LY96	rs11465996	C/G				(−)
				TLR2	rs11938228	C/A				(−)
					rs4696480	A/T				(−)
**van den Reek et al.** [83]	2017	Netherlands	ADA and ETN	CD84	rs6427528	GA	282 ^ϕ^	3 months	Change in PASI	(+)
				HLA-C	HLA-Cw*06	Positive				(−) ^δ^
**Ovejero-Benito et al.** [92]	2017	Spain	ETN	HLA-B/MICA	rs13437088	TT	78	3 months	PASI75	(+)
				MAP3K1	rs96844	CT-CC				(+)
				PTTG1	rs2431697	CT-CC				(+)
				ZNF816A	rs9304742	CC				(+)
				IL12B	rs2546890	AG-GG		6 months		(−)
				GBP6	rs928655	AG-GG				(+)
**Coto-Segura et al.** [136]	2016	Spain	IFX, ADA, and ETN	CARD14	rs11652075	CC	116	24 weeks	PASI75	(+)
				LCE3	indel	Ins				(+)
**Nishikawa et al. ^γ^** [110]	2016	Japan	IFX and ADA	SPEN	rs6701290	G	65	12 weeks	PASI75	(−)
				JAG2	rs3784240	A				(−)
				MACC1	rs2390256	A				(−)
				GUCY1B3	rs2219538	A				(−)
				PDE6A	rs10515637	G				(−)
				CDH23	rs10823825	G				(−)
				SHOC2	rs1927159	A				(+)
				LOC728724	rs7820834	A				(−)
				ADRA2A	rs553668	A				(+)
				KCNIP1	rs4867965	C				(−)
**Mendrinou et al.** [107]	2016	Greek	IFX, ADA, and ETN	FCGR3A	rs396991	G	100	6 months	PASI75	(+) ^ε^
**Masouri et al.** [90]	2016	Greece	IFX, ADA, and ETN	HLA-C	rs10484554	C	228	6 months	PASI75	(+)
				TRAF3IP2	rs13190932	G				(+) ^λ^
				TNFAIP3	rs610604	A				(+) ^ε^
				HLA-A	rs9260313	T				(+) ^δ^
**Coto-Segura et al.** [137]	2015	Spain	IFX, ADA, and ETN	CDKAL12	rs6908435	CC	116	24 weeks	PASI75	(+)
**Batalla et al.** [108]	2015	Spain	IFX, ADA, and ETN	FCGR3A	rs396991	FF	115	6 months	PASI75	(+) ^ε^
**Prieto-Pérez et al.** [131]	2015	Spain	IFX, ADA, and ETN	IL17F	rs763780	CT	180	28 weeks	PASI75	(−) ^δ^/(+) ^λ^
**De Simone et al.** [96]	2015	Italy	ETN	TNF-alfa	rs361525 (-238)	GG	97	12 weeks	PASI75	(+)
					rs1800629 (-308)	GG				(+)
**Batalla et al.** [80]	2015	Spain	anti-TNF	LCE3C_LCE3B	indel	Del	116	24 weeks	PASI75	(−)
**Julià et al.** [138]	2015	Spain	IFX, ADA, and ETN	PDE3A-SLCO1C1	rs3794271	G	130	12 weeks	Change in PASI	(+)
**González-Lara et al.** [101]	2015	Spain	IFX, ADA, ETN, and UTK	TNFRSFB1	rs1061622	G	90	24 weeks	PASI75	(−)
**Gallo et al.** [81]	2013	Spain	IFX, ADA, and ETN	HLA-C	HLA-Cw*06	Positive	109	6 months	PASI75	(−)
				TNF-alfa	rs361525 (-238)	GG				(+)
					rs1799724 (-857)	CT/TT				(+)
					rs1799964 (-1031)	TT				(+)
				IL23R	rs11209026	GG				(+)
**Julià et al.** [105]	2013	Spain	IFX, ADA, and ETN	FCGR2A	rs1801274 (H131R)	HH	70	6 weeks	change in BSA	(+)
				FCGR3A	rs396991 (V158F)	VV				(+)
**Vasilopoulos et al.** [95]	2012	Greek	IFX, ADA, and ETN	TNFA	rs1799724 (-857)	C	80	6 months	PASI75	(+) ^ε^
				TNFRSF1B	rs1061622	T				(+)
**Tejasvi et al.** [103]	2012	USA and Canada	IFX, ADA, and ETN	TNFAIP3	rs610604	G	632	Not specified	Self-evaluated/PASI50 *	(+) ^ε^
					rs2230926/rs610604 (haplotype analysis)	TG				(+)
**Di Renzo et al.** [139]	2012	Italy	IFX, ADA, and ETN	IL-6	rs1800795 (-174)	C	80	24 weeks	PASI75	(+)

Only studies that found statistically significant associations are included; ADA: adalimumab; CVN: copy number variation; ETN: etanercept; EQ-VAS: European Quality of Life Visual Analog Scale; IFX: infliximab; PASI: Psoriasis Area and Severity Index; ^γ^: genome-wide association study (GWAS); ^ε^: results for etanercept; ^δ^: results for adalimumab; ^λ^: results for Infliximab; ^ϕ^: treatment episodes or cycles (one patient could receive more than one cycle of each drug); * self-evaluated using a 0–5 visual analog scale; good if scored from 3 to 5 and poor if scored 0 to 2; PASI50 Toronto cohort; (+): better response; (−) worse response.

The only GWAS evaluating the association between different genetic variants and the response to ustekinumab was recently published. It involved 439 European-descent psoriasis patients that had participated in at least one of the following randomized clinical trials: PHOENIX I [117], PHOENIX II [118], and ACCEPT [126]. SNP rs35569429, which is located on chromosome 4, was significantly associated with ustekinumab, effectivity (change in the PASI at week 12) exceeding the genome-wide significance threshold (beta = −15.83, *p* = 2.42 × 10^−9^). Specifically, patients carrying at least one deletion allele showed a worse response to the drug. This genetic variant is located in an intergenic region upstream of *WDR1*, whose protein regulates immune cell interactions and cell motility. The authors hypothesized that this variation may be involved in promoter/enhancer activities of proximal genes, but the functional effects of this variant still remain largely unknown. Interestingly, patients simultaneously carrying HLA-Cw*06 and the rs35569429-GG genotype presented the highest response to ustekinumab (84.4% achieved PASI75 at week 12), which was significantly higher than that obtained by patients harboring the HLA-Cw*06-negative/rs35569429-GG genotype (71.6%), the HLA-Cw*06-positive/rs35569429-deletion allele (65.5%), and the HLA-Cw*06-negative/rs35569429-deletion allele (38.8%) [140]. 

In this regard, Galluzo et al. also found that the *IL12B* rs6887695-GG genotype and the absence of the *IL12B* rs3212227-AA genotype predicted a better and a longer-lasting response only in patients that simultaneously carried the HLA-Cw*06 allele [124]. Morelli et al. further explored this question using their cohort of 152 Italian patients with psoriasis. As well as identifying different single SNPs associated with different responses to ustekinumab, the investigators found that HLA-Cw*06-positive and HLA-Cw*06-negative patients harbored distinct patterns of SNPs associated with varying clinical responses. Indeed, HLA-Cw*06-positive patients with an optimal response to ustekinumab were characterized by the co-presence of allelic variants of *CDSN*(rs33941312), *PSORS1C3*(rs1265181), *CCHCR1*(rs2073719, rs746647, and rs10484554), *HCP5*(rs2395029), the LCE3A-B intergenic region(rs12030223 and rs6701730), the and HLA-C promoter region(rs13207315 and rs13191343). Concerning SNPs that characterized HLA-Cw*06-negative patients, those present in *MICA*(rs2523497) and *CDSN*(rs1042127 and rs4713436) were associated with a worse response. The authors hypothesized that multi-gene markers, instead of the classical single-SNP biomarker approach, could gain importance in the coming years for informing clinical decisions [141].

**Table 2 ijms-24-09850-t002:** Pharmacogenetic and pharmacogenomic studies on ustekinumab.

Author	Year	Country	Gen	Allele/SNP	Responsive Allele or Genotype	*n*	Follow-Up	Outcome	Response
**Morelli et al.** [141]	2022	Italy	CCHCR1	rs2073719	Not provided	152	12, 28, 64, 76, and 88 weeks	PASI90	(+)
			TNFA	rs1800610			64, 76, 88, and 100 weeks	PASI100	(−)
			Intergenic region upstream of HLA-C	rs12189871 (HLA-Cw*06_LD1)			12, 28, 76, and 88 weeks	PASI90	(+)
				rs4406273 (HLA-Cw*06_LD3)			12, 28, 76, and 88 weeks	PASI90	(+)
				rs9348862			12, 28, 40, and 52	PASI90	(−)
				rs9368670			12, 28, 40, and 52	PASI90	(−)
			PSORS1C3	rs1265181			12, 28, 52, 76, and 88 weeks	PASI90	(+)
			MICA	rs2523497			64, 76, 88, and 100 weeks	PASI100	(−)
			Intergenic region between LCE3B and LCE3A	rs12030223			12, 40,52, and 64 weeks	PASI100	(+)
				rs6701730			12, 40, 52, and 64 weeks	PASI100	(+)
			CDSN	rs1042127			52, 64, 88, and 100 weeks	PASI100	(−)
				rs4713436			52, 64, 88, and 100 weeks	PASI100	(−)
**Connell et al.** [140] **^γ^**	2022	Europe	Intergenic region upstream WDR1	rs35569429	Deletion allele	439	12 weeks	Change in PASI	(−)
**Dand et al.** [82]	2019	UK and Ireland	HLA-C	HLA-Cw*06	Positive	487	6 months	PASI90	(+)
**Loft et al.** [130]	2018	Denmark	IL1B	rs1143623	G/C	230	12 weeks	PASI reduction	(−)
				rs1143627	T/C				(−)
			TIRAP	rs8177374	C/T				(+)
			TLR5	rs5744174	T/C				(+)
**Prieto-Pérez et al.** [132]	2017	Spain	AGBL4	rs191190	TT	69	16 weeks	PASI75	(−)
			HTR2A	rs6311	TT				(−)
			NFKB1A	rs2145623	CC				(−)
			ADAM33	rs2787094	CC				(−)
			IL13	rs848	TT				(−)
			CHUK	rs11591741	GC				(+)
			C17orf51	rs1975974	AG				(+)
			ZNF816A	rs9304742	CT				(+)
			STAT4	rs7574865	GT				(+)
			SLCC22A4	rs1050152	CT				(+)
			C9orf72	rs774359	CT				(+)
**van den Reek et al.** [83]	2017	Netherlands	IL12B	rs3213094	CT	66 ^ϕ^	3 months	Change in PASI	(+)
			TNFAIP3	rs610604	GG			Change in PASI	(−)
**Raposo et al.** [125]	2017	Portugal	HLA-C	HLA-Cw*06	Positive	116	12 and 24	PASI75	(+)
**Talamonti et al.** [122]	2017	Italy	HLA-C	HLA-Cw*06	Positive	255	4, 12, 28, 40, and 52	PASI50, PASI75, PASI90 and PASI100	(+)
**Talamonti et al.** [121]	2016	Italy	HLA-C	HLA-Cw*06	Positive	134	12, 28, 52, and104	PASI75	(+)
**Galluzo et al.** [124]	2016	Italy	HLA-C	HLA-Cw*06	Positive	64	4, 28, 40, and 52	PASI75	(+)
**Li et al.** [127]	2016	USA	HLA-C	HLA-Cw*96	Positive	332	4 and 12 weeks	PASI50 and PASI75	(+)
**Masouri et al.** [90]	2016	Greece	ERAP1	rs151823	CC	22	24 weeks	PASI75	(+)
				rs26653	GG				(+)
**Prieto-Pérez et al.** [131]	2015	Spain	IL-17F	rs763780	TC	70	12 and 24 weeks	PASI75	(−)
**Chiu et al.** [123]	2014	China	HLA-C	HLA-Cw*06	Positive	66	4 and 28 weeks	PASI75 and PASI90	(+)
**Talamonti et al.** [120]	2013	Italy	HLA-C	HLA-Cw*06	Positive	51	4, 12, 28, and 40 weeks	PASI75	(+)

Only studies that found statistically significant associations are included; PASI: Psoriasis Area and Severity Index; ^γ^: genome-wide association study (GWAS); ^ϕ^: treatment episodes or cycles (one patient could receive more than one cycle of each drug); (+): better response; (−) worse response.

#### 3.5.3. Other Biologics

A recent network meta-analysis involving 167 studies and 58.912 patients with psoriasis analyzed the efficacy of systemic pharmacological treatments for chronic plaque psoriasis, including anti-TNF-α and cytokine inhibitors. Anti-IL17 and anti-IL-23 biologics showed a higher proportion of patients achieving PASI90 compared to all the interventions. These results are in line with other meta-analyses that addressed this issue [142,143]. However, despite being the most effective drugs to date, there have been few studies exploring the influence of genetic background on their efficacy. All of them have focused on the pharmacogenetics of anti-IL17 and anti-IL17R drugs. 

The influence of HLA-Cw*06 on secukinumab response was analyzed in the SUPREME study, a 24-week phase IIIb trial that included 434 patients with moderate-to-severe psoriasis. No differences were found in either PASI90 or absolute PASI at 16 and 24 weeks of treatment between HLA-Cw*06-positive and -negative patients. There were also no differences in effectiveness and safety in the open-label extension of the trial (*n* = 384), in which the proportion of patients that reached PASI75/PASI90/PASI100 was comparable across both groups throughout the 72 weeks of treatment [144,145]. A study involving 18 Swiss patients also failed to find differences in secukinumab response based on HLA-Cw*06 status, although the sample size was not powered to identify differences similar to those observed in ustekinumab works [146]. 

On the other side, in a retrospective study involving 151 Italian patients with moderate-to-severe psoriasis treated with secukinumab, HLA-Cw*06 predicted a higher PASI90 achievement from week 16 to week 72. However, this association lost significance when other variables were considered in multivariable models [147]. Another study by Morelli et al. examined 62 Italian patients receiving secukinumab and also observed that HLA-Cw*06-positive patients were more likely to achieve PASI90 at weeks 24, 40, and 56, and PASI100 at weeks 8, 16, and 24 compared to HLA-Cw*06-negative patients. The authors explored the impact of 417 genetic variants previously associated with psoriasis risk or response to biologics. Apart from HLA-Cw*06, different SNPs located in the *HLA-C* promoter region and upstream *HLA-C* were associated with better a response to this drug. The authors hypothesized that these non-coding genetic variants could regulate *HLA-C* transcription. The absence of rs9267325 SNP in *MICB-DT* and the presence of rs909253 and rs1800683 in *LTA* also predicted a stronger response to secukinumab. Interestingly, rs34085293 in *DDX58* and rs2304255 in *TYK2* may identify a subgroup of super-responder patients, as a high proportion of patients carrying these alleles achieved PASI100 both at short- and long-term follow-ups. Both *TYK2* and *DDX58* encode proteins recently involved in the IL-23/IL-17 axis by inducing IL-23 and regulating IL-23-mediated pathways. Interestingly, as with ustekinumab, the authors divided the patients into clusters according to HLA-Cw*06 status and response to secukinumab. Certain SNPs in *DDX58*, the upstream region of *HLA-C*, the *HLA-C* promoter region, or *CCHR1* significantly clustered with HLA-Cw*06 and identified a subgroup of HLA-Cw*06 carriers with a better response. Conversely, different polymorphisms in *MICB-DT, ERAP1,* and *MICA* might identify a subgroup with low response among HLA-Cw*06-negative patients [148]. 

Finally, van Vugt et al. investigated the effect of *IL-17A* gene polymorphisms on secukinumab and ixekizumab response in a cohort of 134 Dutch patients with psoriasis. Although five SNPs in non-coding regions (rs2275913, rs8193037, rs3819025, rs7747909, and rs3748067) were identified, none of them were associated with drug response when evaluating change in PASI or PASI75/90 achievement at 12 and 24 weeks [149]. 

## 4. Conclusions and Future Directions

As reviewed in this article, more than a hundred papers have addressed the influence of different genetic variants on the response to most of the treatments that currently compose the therapeutic arsenal for psoriasis, from topical treatments to biologic drugs. However, despite this huge effort to identify genetic biomarkers that help predict drug response, these biomarkers have barely reached daily clinical practice. Most of the findings of these studies have not been replicated in other series, which are generally of small sample size and mainly include Caucasian and, to a lesser extent, Asian patients. Indeed, the association between different genetic polymorphisms and drug response varies among different populations, which may indicate the existence of population-specific genetic biomarkers. Further research with larger cohorts of patients and including different ethnicities and races will be needed in the coming years to fully establish the role of these polymorphisms on a daily-clinical basis. 

The most widely studied drugs are the anti-TNF-α. While some polymorphisms in the *TNF-α* and *TNFRSF1B* gene have been associated with a better response to these drugs using meta-analyses, these generally include few studies, and the evidence remains conflicting. Examining the response to these drugs as a homogeneous group may have contributed to the heterogeneity in results. As their mechanism of action differs, the polymorphisms evaluated may not exert the same effect with different biologics. Indeed, some studies have found different results when stratifying for etanercept, adalimumab, or infliximab [81,95]. The influence of the HLA-Cw*06 allele on the response to ustekinumab appears to be more robust. Nonetheless, given the high rate of response to the drug regardless of the presence of the allele, some authors question the practical usefulness of its determination. Conducting pharmaco-economic studies could shed light on this issue [150]. Furthermore, pharmacogenetic and pharmacoeconomic studies aimed not only at predicting effectiveness but also at predicting toxicity or effective drug optimization could also provide valuable information on treatment selection. To date, very few studies have addressed this issue.

Finally, epigenetic modifications have also been shown to explain inter-individual differences in response to therapy [151] and have also been involved in psoriasis development [152]. A Spanish research group found that differences in DNA and histone methylation could influence the anti-TNF-α response [153,154]. This largely unexplored genetic field could also provide important clues for identifying predictors for response in this complex heterogeneous disease.

## 5. Limitations

The limitations of the present review include its narrative design. In addition, the search strategy may have limited the scope of this study, as the search terms did not include drugs that are rarely used in our setting but may be frequently used in other regions, such as etretinate or retinoids other than acitretin. This could have made it difficult for us to find studies evaluating the effect of genetic variations in the response to these drugs.

## Figures and Tables

**Figure 1 ijms-24-09850-f001:**
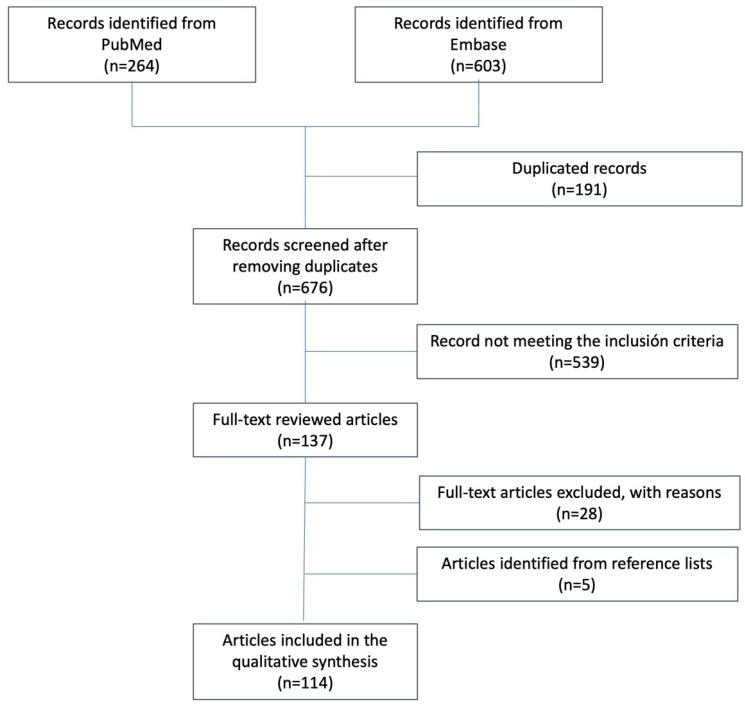
Flow chart showing the study selection process used in this narrative review.

## Data Availability

The data that support the findings of this study are available from the corresponding author upon reasonable request. The data are not publicly available due to privacy or ethical restrictions.

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
