# Peer review of "Genetic Influence on Treatment Response in Psoriasis: New Insights into Personalized Medicine"

_ijms, 2023, doi:10.3390/ijms24129850_

Round 1

Reviewer 1 Report

Both the abstract and the introduction is concise. They clearly indicate the problem of the paper – analysis of available literature in the area of genetic studies in patients with psoriasis treated with different therapies - a review. The introduction is concise and indicates the validity of the analyzes performed. Please add the latest WHO reports (Global Burden of Disease) about psoriasis. I have also found newer articles about extracutaneous manifestations, i.a.: Santus, P. et al. Psoriasis and respiratory comorbidities: The added value of fraction of exhaled nitric oxide as a new method to detect, evaluate, and monitor psoriatic systemic involvement and therapeutic efficacy. BioMed Res. Int. 2018 or Conic, R. et al.. Psoriasis and psoriatic arthritis cardiovascular disease endotypes identified by red blood cell distribution width and mean platelet volume. J. Clin. Med. 2020. In Introduction (lines 54-61) I missed information about neutrophils, which also play an important role in this network of relationships.
The chapters are also thoughtful. Authors start from the description of the literature search process (one hundred and fourteen articles) they have illustrated in the figure. Flow chart of study selection is very helpful. Next, authors divided the studies dependent on therapy: Topical therapies , Phototherapy, Methotrexate, Cyclosporin Acitretin, Apremilast, Anti-TNF-Drugs, Ustekinumab and Others. All the most important conclusions from the research were also presented by the authors in tables that greatly facilitate the reception of the entire work. In my opinion, the work is well organized. It is an excellent summary and introduction for all research scientists who plan research in the field of genetics, e.g. GWAS.
In references part (144 sources) I found above 25% positions from last 3 years.
The conclusion show that we still have a many "to do" to know drug response which is significantly different between populations (it is good that the authors took this into account) but also shows huge inter-individual differences, which only confirms the importance of genetic testing in patients with psoriasis and their families. A review shows why a significant percentage of patients still do not achieve adequate disease control, which is especially important nowadays, because most general therapies are now biologics drugs.

Author Response

Reviewer 1

Both the abstract and the introduction is concise. They clearly indicate the problem of the paper – analysis of available literature in the area of genetic studies in patients with psoriasis treated with different therapies - a review. The introduction is concise and indicates the validity of the analyzes performed. Please add the latest WHO reports (Global Burden of Disease) about psoriasis. I have also found newer articles about extracutaneous manifestations, i.a.: Santus, P. et al. Psoriasis and respiratory comorbidities: The added value of fraction of exhaled nitric oxide as a new method to detect, evaluate, and monitor psoriatic systemic involvement and therapeutic efficacy. BioMed Res. Int. 2018 or Conic, R. et al.. Psoriasis and psoriatic arthritis cardiovascular disease endotypes identified by red blood cell distribution width and mean platelet volume. J. Clin. Med. 2020. 

Response

Firstly, we would like to thank the reviewer for the effort in reviewing our work and for his or her kind words about it. We are pleased that the work is considered of interest. Following the recommendations provided, we have added the latest Global Burden of Disease data about psoriasis (reference 3, lines 39-41). Respiratory comorbidities have been also added to the extracutaneous manifestations of psoriasis (line 46). Although the identification of systemic inflammation through different methods is beyond the scope of the review, we believe that the articles provided by the reviewer are a good example of the systemic nature of psoriasis and have added these citations to the bibliography (references 5 and 6). 

I missed information about neutrophils, which also play an important role in this network of relationships.

Response

We thank the reviewer for this comment. We focus the brief description of the etiopathogenesis of psoriasis on the immune pathways that are altered in this disease (Th1 and Th17), as these are targeted by biologic drugs, which are the drugs most extensively studied. In any case, since, as the reviewer rightly points out, neutrophils play an important role in psoriasis, we have added a brief mention of them (lines 67-68)

The chapters are also thoughtful. Authors start from the description of the literature search process (one hundred and fourteen articles) they have illustrated in the figure. Flow chart of study selection is very helpful. Next, authors divided the studies dependent on therapy: Topical therapies , Phototherapy, Methotrexate, Cyclosporin Acitretin, Apremilast, Anti-TNF-Drugs, Ustekinumab and Others. All the most important conclusions from the research were also presented by the authors in tables that greatly facilitate the reception of the entire work. In my opinion, the work is well organized. It is an excellent summary and introduction for all research scientists who plan research in the field of genetics, e.g. GWAS.
In references part (144 sources) I found above 25% positions from last 3 years.
The conclusion show that we still have a many "to do" to know drug response which is significantly different between populations (it is good that the authors took this into account) but also shows huge inter-individual differences, which only confirms the importance of genetic testing in patients with psoriasis and their families. A review shows why a significant percentage of patients still do not achieve adequate disease control, which is especially important nowadays, because most general therapies are now biologics drugs.

Response

We are very grateful for the thoughtful consideration of our work.

Reviewer 2 Report

The authors reviewed associations between genetic variations and treatment response to psoriasis. The articles was comprehensive and was well-sturcutured. I have serveral suggestions for the authors.

1. The literature search strategy limited the scope of the study. For example, etretinate instead of acitretin is used for psoriasis treatment in Japan, and therefore it might be possible that some studies investigating associations between etrtinate or other retinoids and genetic variations were not included in the present study. This limitation should be mentioned.

2. For reference 47 and other single-center study, it is suggested to mention the ethnic group as this information is important for genetic study.

3. In the main text, the terms cyclosporin and ciclosporin were used alternatively. It is suggested to use the same term for this medication.

Author Response

The authors reviewed associations between genetic variations and treatment response to psoriasis. The articles was comprehensive and was well-sturcutured. I have serveral suggestions for the authors.

  1. The literature search strategy limited the scope of the study. For example, etretinate instead of acitretin is used for psoriasis treatment in Japan, and therefore it might be possible that some studies investigating associations between etrtinate or other retinoids and genetic variations were not included in the present study. This limitation should be mentioned.

Response

We very much appreciate the reviewer's words and the effort to review our work. We agree that the search criteria may have limited the scope of the study, especially in the case of etretinate, which is not used in our setting. We have added a new section called “Limitations” at the end of the current version of the manuscript (lines 72-76).

  1. For reference 47 and other single-center study, it is suggested to mention the ethnic group as this information is important for genetic study.

Response

The reviewer makes a very interesting point. Ethnicity is in many of these studies difficult to obtain as there is great heterogeneity in data reporting. However, following your recommendation, we have added the country in which these studies have been carried out, which also provides valuable information for the reader.

Response

  1. In the main text, the terms cyclosporin and ciclosporin were used alternatively. It is suggested to use the same term for this medication.

Response

We thank the reviewer for this comment and we apologize for the mistake. To unify terms, we use cyclosporine. We have corrected this issue in the current version of the manuscript.